# A Variety of *Leptospira* Serovar Distribution in Bullfighting Cattle in Southern of Thailand

Ruttayaporn Ngasaman [1,*], Baramee Chanchayanon [1], Domechai Kaewnoi [1] and Ketsarin Kamyingkird [2]

[1] Faculty of Veterinary Science, Prince of Sonkla University, Kanjanavanich Road, Korhong, Hatyai, Songkhla 90110, Thailand; baramee.c@psu.ac.th (B.C.); domechai.k@psu.ac.th (D.K.)
[2] Department of Parasitology, Faculty of Veterinary Medicine, Kasetsart University, Ladyao, Chatuchak, Bangkok 10900, Thailand; ketsarin.ka@ku.th
[*] Correspondence: ruttayaporn.n@psu.ac.th

**Simple Summary:** This is the first report of seroprevalence and molecular detection of leptospirosis in bullfighting cattle in Southern Thailand. Bullfighting cattle are famous for their fighting game that has attracted many audiences to arenas. Leptospirosis is an endemic disease in Thailand of both humans and many species of animals; however, this disease has never been investigated in bullfighting. Therefore, we aimed to determine leptospirosis status in bullfighting cattle in order to prevent disease transmission to the environment and humans by treating infected bulls and vaccinating all bulls on the farm. This study collected 200 blood samples from bulls that visited an animal hospital for a health check before a fighting game. A lepto-latex test was used for screening the antibody against pathogenic *Leptospira* in the serum, then all positive samples were sent for MAT analysis at the National Institute of Animal Health, Department of Livestock, Thailand. At the same time, all samples were analysed for the presence of *Leptospira* DNA using LipL32 PCR. The results from this study showed seroprevalence of *Leptospira* of 27% and 13% of DNA detection. Antibodies directed against nine different serovars were detected in the bullfighting cattle; the most common were Ranarum and Shermani. The results in this study showed that bullfighting cattle could be an important reservoir of leptospirosis in Southern Thailand.

**Abstract:** Southern of Thailand is an endemic area for human leptospirosis, and many species of animals act as a reservoir. Therefore, this study aimed to determine leptospirosis in bullfighting cattle by detecting antibodies against *Leptospira* using molecular detection. A total of 200 serum samples were screened using a lepto-latex test and then subjected to a microagglutination test. DNA was analyzed using polymerase chain reaction for the *LipL32* gene of pathogenic *Leptospira*. Screening tests identified 127 positive samples (63.50%). The seroprevalence detected by MAT was 27.00%, while molecular detection showed a prevalence of active infection of 13%. Nine serovars of *Leptospira interrogans*—Ranarum, Shermani, Pyrogenes, Bratislava, Pomona, Autumnalis, Habdomadis, Louisiana, and Bataviae—were identified. Ranarum and Shermani were the main serovars circulating in the bullfighting cattle. A total of 96.29% (52/54) of the MAT-positive showed a multi-serovar infection pattern. The pattern of serovars Ranarum–Shermani was the most common finding (64.81%) followed by Pyogenes–Ranarum–Shermani (11.11%), Autumnalis–Ranarum–Shermani (7.41%), and Bratislava–Louisiana–Pomona–Pyogenes–Ranarum–Shermani (3.70%). As a result, a relatively high seroprevalence showed that bulls might have a high chance of infection in the past, while detection of DNA meant that some bulls were in an active infection period. Hence, the bulls might play an important role in disease transmission to the environment, animals, and humans during their infection. The variety of serovars found might indicate many origins of pathogens or multiple infections. The application of disease surveillance will reduce the percentage of carriers in bulls, and might reduce the risk to human health.

**Keywords:** bullfighting cattle; carrier status; endemic area; leptospirosis; Thailand

## 1. Introduction

Bullfighting cattle are selected from domestic cattle (*Bos indicus*) that demonstrate good performance in Thai bullfighting without a matador. Thai bullfighting is a very popular event in the southern region, and can be staged in a stadium or simply on cleared, open farmland. Normally, bulls aged 2–5 years are used for fighting. Bull health is checked before a performance, but zoonoses are not assessed. *Leptospira* spp. is the causative agent of leptospirosis and is distributed worldwide, especially in tropical areas, including Thailand [1]. Rodents are the main reservoir, and can carry different pathogenic serovars of *Leptospira* spp. capable of causing disease in humans and animals [2]. In Thailand, *Leptospira* has been identified in a wide range of reservoir hosts, including pets such as dogs and cats, and in livestock such as pigs, cattle, buffaloes and goats [3–7]. It has also been identified in wild animals such as monkeys [8]. Infected animals, which once infected, may shed the organism in their urine intermittently or continuously throughout life, resulting in contamination of the environment, particularly water [9]. The general mode of transmission is indirect contact with contaminated water during the rainy season and after flooding [10].

Human leptospirosis mostly occurs in the northeastern and southern regions of Thailand; the southern region is classified as an important epidemic area (www.boe.moph.go.th, accessed on 9 January 2022). Pathogenic *Leptospira* causes bovine leptospirosis. The host-adapted *Leptospira* serovar in cattle, which can be infected at any age, is *Leptospira hardjo-bovis.* However, cattle may accidentally be infected with other non-host-adapted serovars such as *Leptospira interrogans Pomona, Icterohaemorrhagiae, Canicola, and Grippotyphosa* (www.thecattlesite.com, accessed on 9 January 2022). Although the seroprevalence of leptospirosis in cattle of Thailand is not high, *Leptospira* can persist in the reproductive tract for a long time, and can cause chronic bovine leptospirosis with signs of infertility [7]. The clinical signs in calves include fever, anemia, red urine, jaundice, and sometimes death in three to five days. In older cattle, the initial symptoms such as fever and lethargy are often milder and usually go unnoticed, but reduced milk yield in lactating cows may lead to economic losses for farmers.

Although bullfighting cattle in Southern Thailand are mainly raised for fighting, they can also be consumed. In addition to economic loss, leptospirosis causes problems for human health in Thailand. As a previous study reported, the incidence of human leptospirosis in Thailand is 6.6 per 100,000 members of the population [11]. It has been found that humans working with animals or environmental water have a high chance of infection; i.e., 4.31 times more than other occupations without contact with environmental water [12]. Therefore, this study aimed to determine the carrier status of leptospirosis, survey the circulating serovars, and assess the current infection burden in bullfighting cattle. From the perspective of human health, this study was expected to survey the contaminated environmental source of *Leptospira* shed from bullfighting cattle that were raised as pets for a fighting game. As those bulls were in a closed relationship with humans, the results of this research may raise awareness and help to prevent disease transmission to humans.

## 2. Materials and Methods

### 2.1. Sampling and Sample Preparation

The sample size was calculated based on the prevalence of *Leptospira* infection in domestic cattle of about 28.1% [13]. The sample size was calculated by using a free online program [14] with an estimated number of bullfighting cattle of 20,000, a desired precision of 10%, and a 95% confidence interval. The sample size was 78 samples. However, this study was designed to collect 200 blood samples from bullfighting cattle aged 2–5 years in Southern Thailand during visits to the animal hospital of the Faculty of Veterinary Science between 2017–2019. Those bulls showed no clinical signs during their visits, but their owners complained that they were exercise-intolerant. The samples with and without coagulant were stored at 4 °C until analysis. Serum was collected from coagulated tubes and stored at −20 °C until proceeding with serological tests. Whole blood in EDTA tubes was used for DNA extraction with a commercial kit (QIAamp® genomic DNA and RNA

kits) according to the manufacturer's instructions. The purified DNA was stored at −20 °C for specific polymerase chain reaction (PCR) analysis.

### 2.2. Serological Analysis

A screening test was performed using the lepto-latex test, which was developed by the Department of Medical Sciences, Ministry of Public Health, Thailand. Briefly, 10 µL of serum was mixed with 10 µL of *Leptospira* antigen (*L. interrogans* Pyrogenes) on a glass slide with a black background, mixed gently using a sterile tip, and then observed by eye after 2–5 min. Positive results showed agglutination. After that, positive sera from the lepto-latex test were sent for serovar analysis using a microagglutination test (MAT) at the National Institute of Animal Health Thailand (NIAH). The cut-off titer used to evaluate antibodies against *Leptospira* serovars was ≥1:100.

### 2.3. Molecular Detection

Conventional PCR to detect the *LipL32* gene of *Leptospira* was performed as previously described [14]; it identified fragments of 423 base pairs in size. Each 12 µL reaction included 5.0 µL of Taq polymerase master mix (KAPA®, Tokyo, Japan), 0.25 µL of each primer, 4.0 µL of ultrapure water, and 2.0 µL of DNA template. The PCR cycle consisted of an initial denaturation step at 95 °C for 5 min, followed by 35 cycles of 1 min at 94 °C for denaturation, 1 min at 60 °C for annealing, and 1 min at 72 °C for extension, with a final extension at 72 °C for 7 min. The PCR products were electrophoresed on 2% agarose gels at 100 V for 30 min.

### 2.4. Statistical Analysis

Raw data were analyzed as a percentage using Microsoft Excel (Microsoft, Redmond, WA, USA). The proportions and 95% confidence intervals were calculated using the website https://sample-size.net (accessed on 9 January 2022). The associations among the three tests—lepto-latex, MAT, and PCR—were anlyzed by using Fisher's exact test in a free online program (https://www.socscistatistics.com, accessed on 24 April 2022).

## 3. Results

A total of 200 samples screened using the lepto-latex test showed 127 positive samples with an agglutination score of +1 to +4; 73 samples showed negative results. The positive samples for each agglutination score were 73, 37, 14, and 3 samples, respectively. According to the MAT analysis, 54 samples had a positive titer at >1:100; 37, 14, and 3 samples, respectively, for agglutination scores of +2, +3, and +4. Molecular detection using PCR targeting the LipL32 gene identified 26 positive samples with an agglutination score of 0 to +3. The number of positive samples in the non-detectable group (score 0) to agglutination score +3 were 10, 10, 3, and 3, respectively (Table 1).

Serovar identification in 54 positive samples according to the MAT analysis showed that only two samples were a single-serovar infection, one with Ranarum and another with Shermani. A total of 52 samples were multiserovar infections. There were nine serovars identified in this study: Ranarum (RAN), Shermani (SHE), Pyrogenes (PYR), Bratislava (AUS), Pomona (POM), Autumnalis (AUT), Habdomadis (HAB), Louisiana (LOU), and Bataviae (BAT). RAN was the most common serovar in the bullfighting cattle (100%), followed by SHE (96.3%), PYR (16.67%), AUS (12.96%), POM (5.56%), AUT (3.70%), HAB (3.70%), LOU (3.7%), and BAT (1.85%). Moreover, 52 out of 54 samples were identified as having more than two serovars. A total of 35 samples contained the serovars RAN–SHE (64.81%), and 13 samples (24.07%) contained three serovars: six with PYR–RAN–SHE (11.11%), four with AUT–RAN–SHE (7.41%), one with AUS–RAN–SHE (1.85%), one with BAT–RAN–SHE (1.85%), and one with HEB–RAN–SHE (1.85%). Only one sample was identified containing the HEB–PYR–RAN–SHE serovars (1.85%). There was one sample containing five serovars; i.e., AUS–AUT–POM–RAN–SHE (1.85%), and two samples contained six serovars; i.e., AUS–LOU–POM–PYR–RAN–SHE (3.70%) (Table 2). Regarding

the association between results for the lepto-latex test score 0 to score 4 with the MAT and PCR tests, it was found that the results of lepto latex scores +2, +3, and +4 and MAT results were not significantly dependent at $p < 0.05$.

**Table 1.** Positive samples from serological analysis using lepto-latex test, microagglutination test (MAT), and molecular analysis by polymerase chain reaction (PCR) targeting LipL32 gene of pathogenic *Leptospira* spp.

| Lepto-Latex Test | | | MAT | | *LipL 32* PCR | |
|---|---|---|---|---|---|---|
| Score | No. of Samples | % (95% CI) | No. of Positive | % (95% CI) | No. of Positive | % (95% CI) |
| ND | 73 | 36.5 (29.82–43.58) | ND | 0 | 10 | 5.0 (2.42–9.00) |
| +1 | 73 | 36.5 (29.82–43.58) | ND | 0 | 10 | 5.0 (2.42–9.00) |
| +2 | 37 | 18.5 (13.37–24.59) | 37 | 18.5 (13.37–24.59) | 3 | 1.5 (0.31–4.32) |
| +3 | 14 | 7.0 (3.88–11.47) | 14 | 7.0 (3.88–11.47) | 3 | 1.5 (0.31–4.32) |
| +4 | 3 | 1.5 (0.31–4.32) | 3 | 1.5 (0.31–4.32) | 0 | 0 |
| Total | 200 | | 54 | 27 (20.98–33.72) | 26 | 13 (8.67–18.47) |

ND: Not detected.

**Table 2.** Serogroup identification by MAT analysis at the titer >1:100 of total 54 positive samples. RAN: Ranarum; SHE: Shermani; AUT: Autumnalis; AUS: Bratislava; BAT: Bataviae; HEB: Habdomadis; PYR: Pyrogenes; POM: Pomona; LOU: Louisiana.

| Serogroups | No. of Positive | % | (95% CI) |
|---|---|---|---|
| RAN | 1 | 1.85 | (0.05–9.89) |
| SHE | 1 | 1.85 | (0.05–9.89) |
| RAN–SHE | 35 | 64.81 | (50.62–77.32) |
| AUT–RAN–SHE | 4 | 7.41 | (2.05–17.89) |
| AUS–RAN–SHE | 1 | 1.85 | (0.05–9.89) |
| BAT–RAN–SHE | 1 | 1.85 | (0.05–9.89) |
| HEB–RAN–SHE | 1 | 1.85 | (0.05–9.89 |
| PYR–RAN–SHE | 6 | 11.11 | (4.19–22.63) |
| HEB–PYR–RAN–SHE | 1 | 1.85 | (0.05–9.89) |
| AUS–AUT–POM–RAN–SHE | 1 | 1.85 | (0.05–9.89) |
| AUS–LOU–POM–PYR–RAN–SHE | 2 | 3.70 | (0.45–12.75) |
| | 54 | 100.00 | |

## 4. Discussion

Most of the identifications by MAT were associated with a late stage of infection with agglutination score +2 (18.5%), followed by agglutination score +3 (7%) and agglutination score +4 (1.5%), according to the screening test. At the early stages of infection, agglutination was very weak (+1) and could not be detected by MAT. However, stronger agglutination (+3 and +4) according to the lepto-latex test had a lower percentage of positive samples by MAT than those with an agglutination score of +2. This might indicate that antibody levels were reduced in the late stage of infection. The total seroprevalence of leptospirosis in the bullfighting cattle according to the MAT results was 27%, similar to a study of domestic cattle in Thailand that showed a seroprevalence of 28.1% [15]. This can be compared with results from the Salakphra wildlife sanctuary in Thailand, where the seroprevalence according to MAT in cattle was 92.2% [16]. This might have been because

the cattle in Salakphra live in wetlands, but bullfighting cattle are always raised in a stable. Since bullfighting cattle have very close contact with people, such a percentage of seroprevalence and leptospiremia status without clinical signs in bullfighting cattle has a strong influence on transmission to humans. Similar to previous reports in Nan province, pathogenic *Leptospira* occur commonly in asymptomatic domestic animals, humans, and environmental water samples, which emphasises the potential for zoonotic transmission in the province [17]. A previous study using the loop-mediated isothermal amplification (LAMP) technique to detect pathogenic leptospiral 16S rDNA in urine samples of domestic cattle and buffalo in Thailand demonstrated a prevalence of 5.90–8.43% [18,19], but this study showed more active infections (13%) in the blood. This might have been because the *LipL32* protein is expressed at high levels during infection [20]. In addition, the sensitivity (98.68%) and specificity (100%) of the *LipL32*-PCR [21] were higher than the sensitivity and specificity of LAMP (96.8% and 97.0%, respectively) [22]. According to the results, the DNA was mostly detected with a lepto-latex test agglutination score of 0–1 and a non-MAT reaction (no antibody). However, six samples identified both DNA and antibodies at lepto-latex test agglutination scores of +2 and +3 (3 samples of each). This might have been due to the bull having a previous antibody and acquiring a new serovar infection during blood sampling. However, the detected antibody against serovars of those six samples were from the previous infection. As a previous study suggested, a rapid diagnosis that could be applied at the point of care was essential for acute infection management, and the method to understand of the idiversity of pathogenic *Leptospira* spp. had to be used later [23]. Therefore, this research used lepto for screening, and also used MAT and PCR, which required special in-house laboratory methods to confirm and understand the circulation of serovars. Among the 13% PCR-positive cattle with leptospirosis in this study, they only showed exercise intolerance. Generally, bullfighting cattle are selected from high-performance indigenous beef cattle. The infected ones only showed lethargy during exercise; this was barely observed during visits to the animal hospital and was considered as mild or as undetected clinical signs. Moreover, those bulls were males, and did not show specific clinical signs of reproductive failure. In addition, being an indigenous native breed led to disease intolerance.

Our results were higher than those of the study in [24], which reported a 6.44% incidence of leptospirosis in dairy cattle in Brazil. However, the animal population in our study were bulls, which only refers to male cattle (because bullfighting cattle are only male beef cattle), not females.

According to the symptoms and effects of leptospirosis in cattle, bovine leptospirosis can be categorized as two syndromes: incidental and adapted bovine leptospirosis [25]. Both of these syndromes result in chronic leptospirosis characterized by reproductive disorders presenting primarily as late-term abortion, the more visible symptom in female cattle [25]. The authors of that study also stated that, 'This silent disease can go unnoticed and undiagnosed, compromising reproductive efficiency with a consequent decrease in the productivity of herds over long periods.' For these reasons, to characterize clinical symptoms of leptospirosis in male cattle in a cross-sectional epidemiological study is difficult because the disease can go unnoticed and undiagnosed in female cattle, as described above.

In addition, incidental bovine leptospirosis is caused by the serovars maintained by the other domestic and free-living animals, while adapted bovine leptospirosis is caused by strains in cattle, and it does not require other animals for transmission [26]. It was previously reported that the Hardjo serovar of *Leptospira* (L.) interrogans was highly associated with bovine genital leptospirosis (BGL), while the Pomona serovar of L. interrogans was associated with BGL at a low level [25]. Our study identified nine serovars of L. interrogans, including; Ranarum, Shermani, Pyrogenes, Bratislava, Pomona, Autumnalis, Habdomadis, Louisiana, and Bataviae. There was no Hardjo serovar detected in our study. Therefore, detection of BGL symptoms in our study was difficult as well.

Regarding the other point, we also mentioned in the Introduction that bovine leptospirosis may cause severe symptoms and death in calves. However, there were cattle

aged 2–5 years that were not considered as calves in this study. Therefore, unnoticed and undiagnosed symptoms in the 2–5-year-old male adult cattle in our study were obviously supported by the reasons mentioned above. Thus, *Leptospira* can be localized and persistent in the kidney and genital tract of the male [27]. However, we did not collect animal urine or detect kidney conditions in this study. Future studies may require measurement of the kidney function and identify the pathogens in the urine of male animals.

Antileptospiral antibodies were identified, including Ranarum, Shermani, Pyrogenes, Bratislava, Pomona, Autumnalis, Habdomadis, Louisiana, and Batavia. Among those, the antileptospiral antibodies Ranarum (54/54) and Shermani (53/54) were the most commonly found. This meant that both serovars were circulating in the bullfighting cattle. This was similar to a previous study that determined that the most predominant serovars in cattle were Shermani and Ranarum [15]. However, another study of cattle in Thailand determined that the most commonly detected serovars were Ranarum, Sejroe, and Mini [7]. This was different from the findings in cattle at the Salakphra wildlife sanctuary, which showed that the most common serovar was Tarassovi [16]. Moreover, the present study showed a high rate of coreaction according to the MAT test, with three to six serovars (31.48%). It might be that there was antibody against those *Leptospira* serovars in the samples. However, Ranarum and Shermani were the main serovars in all patterns of coreaction, which suggested that the cattle were more susceptible to these serovars. The serovars Autumnalis and Bratisalava are commonly found in humans, buffaloes, and pigs [7,15,28]. Therefore, infection in cattle might be associated with spread from humans, buffaloes, and pigs. In the northeastern provinces of Thailand, the most common serovars infecting small wild mammals are Autumnalis and Canicola [29]. The most commonly detected serovars of leptospirosis in rodents and shrews in high and low endemic areas are Pyrogenes (39.1%), Sejroe (19.1%), Bataviae (10.0%), Pomona (6.4%), Autumnalis (5.5%), Copenhageni (3.6%), and Javanica (3.6%) [30]. Serovars Bataviae, Habdomadis, Pyrogenes, Pomona, and Louisiana are not commonly found in livestock in Thailand. However, these serovars are found in livestock in other countries, such as in Sudan, where the serovars Hebdomadis, Bataviae, and Pomona have been found in cattle and sheep [31]. In India, the seropositivity of leptospirosis in cattle was higher than in Thailand; it was 50.85%, with the most common serovar being Icterohaemorrhagiae [32]. In northeastern Malaysia, about 81.7% were seropositive, broader with Thailand, with the most common serovar being Sarawak [33]. The distribution of leptospiral serovars in cattle, goats, and sheep in flood-prone Kelantan, Malaysia included Hardjo-bovis (3.70%), Hebdomadis (2.08%), and Pomona (1.04%) [34]. In an outbreak of human leptospirosis after a flood in Thung Song District, Nakhon Si Thammarat, the dominant serovars were Shermani and Sejroe; dogs, cats, cattle, and rats were suspected to be the primary sources of infection [35]. The serovar Icterohaemorrhagiae was the most common serovar in human cases after river water rafting in Satun province [36].

It was found that the factors for increased risk of severe leptospirosis in humans were associated with living near a rubber tree plantation and bathing in natural bodies of water [37]. Moreover, it was found that flooding strongly contributed to disease transmission and the rate of leptospirosis transmission in a contaminated environment; this was the most important parameter regarding the total number of human cases [38]. The style of bull raising in the southern region normally occurs on or near a rubber tree plantation, and the supplied grass is from a wetland area. Therefore, the bulls have a high chance of coming into contact with contaminated water. The infected bulls did not show specific signs, but they may play an important role in maintaining and shedding many serovars of pathogenic *Leptospira* into the environment and to humans in the future. Due to the small number of samples (200) collected in the animal hospital of the faculty, our results might not present the true prevalence of leptospirosis in the bullfighting cattle population in Southern Thailand. Moreover, our next study should include the causal risk factors of leptospirosis in bullfighting cattle.

## 5. Conclusions

This study revealed that bullfighting cattle are a potential source of many serovars of *Leptospira* spp. This is a serious threat to the environment. Humans have a high risk of being infected, especially farmers, bull keepers, their family members, and the audiences in the fighting arena. Moreover, domestic animals that are raised on the same farm may acquire infections, affecting their health and causing a loss in production. Therefore, this study suggests a surveillance program based on serological and molecular diagnosis of acute and chronic leptospirosis in bullfighting cattle routinely. Treatments need to be applied in the infected cattle. Due to there being no *Leptospira* vaccine program available in Thailand, the department of livestock development should consider establishing a *Leptospira* vaccine for livestock in the future. In addition, people raising bulls for fighting should apply biosecurity on their farms in order to prevent and control this pathogen. A vaccine should be established for bullfighting cattle because they are in closer contact with humans than other domestic cattle.

**Author Contributions:** R.N.: conceptualization, methodology, visualization, writing—original draft, writing—review and editing; B.C.: formal analysis, investigation; D.K.: project administration, resources, software; K.K.: supervision, validation, writing—review and editing. All authors have read and agreed to the published version of the manuscript.

**Funding:** There was no funding to support this research.

**Institutional Review Board Statement:** The blood sampling from animal protocol was used in this study. The animal sampling protocol was approved by the Ethics Committee of Kasetsart University (Approval no. ACKU 01360).

**Informed Consent Statement:** Not applicable.

**Data Availability Statement:** Not applicable.

**Acknowledgments:** Special thanks to the staff of the animal hospital for providing help in collecting samples from the bullfighting cattle. We also thank the faculty of Veterinary Science, Prince of Songkla University, for providing assistance in the use of laboratories in this research.

**Conflicts of Interest:** The authors declare that there are no conflicts of interest.

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
