# Peer review of "A Variety of Leptospira Serovar Distribution in Bullfighting Cattle in Southern of Thailand"

_zoonoticdis, doi:10.3390/zoonoticdis2020008_

Round 1
Reviewer 1 Report
General Comments
The study identifies the risk of contamination of Leptospira and the human population with leptospirosis. It may constitute an important contribution to public health, but it is only of a descriptive nature, it does not give us any causal relationship, it only alerts us to the potential risk of contamination by leptospirosis.
In my opinion, the article would benefit from the addition of inferential analysis to better discriminate the risk and eventual causal factors.
Specific comments:
Introduction:
It addresses the fundamental aspects of the problem under study.
Materials and Methods:
The analysis is exclusively descriptive.
Results: Exclusively descriptive results.
Discussion:
It compares the main findings of this article with those published in other studies.
The authors do not present any limitations of the study.
Conclusions:
Very succinct
Author Response
Response as attached file

Reviewer 2 Report
This study is very interesting as it described an important zoonotic disease. It is important for the molecular study to describe the sequence of each primer used.
Author Response
Response as attached file

Reviewer 3 Report
Dear authors,
I would encourage resubmission after thorough revision of the manuscript. In my opinion, it cannot be published in its present form.
Some specific comments would be:
Usually, I don't feel qualified to judge about the English language and style. In this case, however, I have the impression that a considerable revision of the English language is required - if only to really understand the content of the manuscript.
Abstract: The abstract should start with an introductory sentence (background). PCR studies are mentioned (DNA analysis), but no further information and no results are given. Abstract needs to be completely revised.
PCR studies were performed with serum, but it is known that PCR is positive in serum only in the early phase of infection for about one week. The results of the present study are surprising and one has to consider what this might be due to. Other investigators use urine for similar studies. Would a methodological error in the PCR be conceivable here?
When multiple serovars react in MAT, it is often due to "co-reactions," but one cannot assume that infections with all reacting serovars have occurred. Here, the results should be interpreted and discussed accordingly.
Among others, the recently published review by Sykes et al. should also be considered in this work (https://doi.org/10.3390/pathogens11040395)
Author Response
Response as attached file

Round 2
Reviewer 3 Report
Dear authors,
I agree that Bullfighting in Thailand and the husbandry of the bulls are certainly relevant as a possible source of infection. However, I am still struggling to understand how so many blood samples can be PCR positive in clinically healthy bulls. (Line 48: “Bull health is checked before performance”, but regardless of this there are really 13% positive PCR-results examining the serum samples? Even if the symptoms are mild, by checking health before performance some symptoms are to be expected..? Or how is the health of the bulls checked before performance?) How old were the bulls? How likely is a new infection in cases where both antibodies and PCR-positive results are there at the same time?
Line 13/14 How is it possible to prevent disease transmission to the environment?
Line 20 It cannot be said that 9 serovars circulated, but only that antibodies directed against 9 different serovars were detected
Line 37 and 175: Monkey? I thought there were bulls in the study?
Results: I would recommend to include the tables in the manuscript instead of the supplementary file
Line 163-164: “…pathogenic Leptospira occur commonly in asymptomatic domestic animals, humans and environmental water samples…” – yes, but for spreading of the bacteria urine samples would have to be tested; leptospires persist in the renal tubules, not in the blood..?
Line 182: “The serovars of Leptospira interrogans identified in this study were…” – this should be rephrased: anti-leptospiral antibodies were identified, but not the serovars (Line 184 the same)
Round 3
Reviewer 3 Report
Dear authors,
you have addressed my comments, but I still have difficulties with the PCR-results of the blood samples. Thus, I recommended to get a second opinion here.
Sincerely
Author Response
"Please see the attachment."
